# Study protocol for a pilot randomised controlled trial evaluating the feasibility and effectiveness of non-pharmacological interventions to recover functionality after a transient ischaemic attack or a minor stroke: the 'Back to Normal' trial

Micaela Gonçalves  ,[1,2] Maria João Lima,[3] Ângelo Fonseca,[3] Cristina Duque,[3] Ana Rute Costa,[1,2,4] Vitor Tedim Cruz[1,2,3]

For numbered affiliations see end of article.

**Correspondence to**
Micaela Gonçalves;
micaela.goncalves@ispup.up.pt

## ABSTRACT

**Introduction** Transient ischaemic attack (TIA) and minor stroke are frequently assumed as temporary or non-disabling events. However, evidence suggests that these patients can experience relevant impairment and functional disability. Therefore, the present study aims to evaluate the feasibility and effectiveness of a 3-month multidomain intervention programme, composed of five non-pharmacological strategies, aimed at accelerating return to pre-event level of functionality in patients with TIA or minor stroke.

**Methods and analysis** Patients diagnosed with a TIA or a minor stroke are being recruited at the emergency or neurology departments of the Hospital Pedro Hispano, located in Matosinhos, Portugal (n=70). Those who accept to participate will be randomly allocated to two groups (1:1): (a) Intervention—receives a 3 months combined approach, initiating early post-event, composed of cognitive training, physical exercise, nutrition, psychoeducation and assessment/correction of hearing loss; (b) Control—participants will not be subject to any intervention. Both groups will receive the usual standard of care provided to these diseases. Recruitment began in May 2022 and is expected to continue until March 2023. Socio-demographic characteristics, lifestyles, health status, cognitive function, symptoms of anxiety and depression and quality of life will be assessed; as well as anthropometry, blood pressure and physical condition. Time to complete or partial recovery of instrumental activities of daily living will be assessed using an adapted version of the Frenchay Activities Index. All participants will be evaluated before the intervention and after 3 months.

**Ethics and dissemination** This study was approved by the Ethics Committee of the Local Health Unit of Matosinhos (Ref. 75/CES/JAS). Written informed consent will be required from all the participants; data protection and confidentiality will be also ensured. The findings of this project are expected to be submitted for publication in scientific articles, and the main results will be presented at relevant scientific meetings.

**Trial registration number** NCT05369637.

## STRENGTHS AND LIMITATIONS OF THIS STUDY

⇒ This protocol describes a pilot randomised controlled trial addressing the effect of an innovative multidomain non-pharmacological intervention programme on post-transient ischaemic attack and minor stroke functionality.

⇒ The intervention programme will start early after the diagnosis of the vascular event, maximising health gain and targeting brain plasticity.

⇒ The main outcome is based on the time to recover instrumental activities of daily living after the event, which is more suitable due to the characteristics of residual impairment.

⇒ This study will explore a wide range of different outcomes, including study feasibility, cognitive function, lifestyle behaviours, physical and emotional status.

⇒ Given the characteristics of the intervention, blinding of the participants is not possible. In addition, the main outcome is evaluated by self-report and is based on a predefined list of activities of daily living. Therefore, we cannot exclude the existence of recall bias regarding our main outcome.

## INTRODUCTION

A transient ischaemic attack (TIA) is defined as a 'transient episode of neurological dysfunction caused by focal brain, spinal cord, or retinal ischemia, without acute infarction' by the American Heart Association,[1] while a minor stroke is suggested to occur when a patient has a score of ≤3 in the National Institutes of Health Stroke Scale (NIHSS).[2] Population-based studies have shown that the age-adjusted incidence of TIA in the European population ranges from 28 to 59/100 000/year[3] and, in Portugal, a higher annual TIA incidence of 74/100 00/people was estimated between 2009 and 2011.[4] This is

particularly relevant since previous studies have described an increased risk of stroke after these events,[5] and, in 2019, stroke was the third leading cause of disability-adjusted life years.[6]

The study of the overall morbidity associated with a TIA or a minor stroke is often neglected, since most of the time they are assumed as being transient or non-disabling events. However, evidence suggests that these patients can experience relevant disability even with a NIHSS of 0 or 1.[7] A decreased performance in activities of daily living (ADLs) up to 1-year post-event was also observed, including in patients without any apparent physical disability.[8 9] Other studies showed a high prevalence of depression and apathy,[10] cognitive impairment,[11] physical deficits[12] and hearing degeneration,[13] with implications on patients' return to work, social relations and activities.[14] Moreover, European guidelines on management of TIA recommend for early initiation of secondary prevention following such an event,[15] focusing particularly on the leading risk factors for stroke, namely high systolic blood pressure, body mass index, fasting plasma glucose and smoking.[6]

Taken together, these findings highlight the importance of multidomain interventions aiming to promote functioning levels among patients with TIA or minor stroke, and to develop preventive actions and health promotion plans specifically designed for this group. As such, this pilot study primarily aims to explore the feasibility and the effectiveness of an innovative multidomain intervention programme, composed of five strategies: cognitive training, physical exercise, nutrition education, psychoeducation and assessment/correction of hearing problems. These are aimed at accelerating the return to pre-event functionality in patients with a TIA or a minor stroke diagnosis. The primary objectives are as follows:

1. To assess the feasibility of methodology for early recruitment, collecting information and delivering the interventions.
2. To estimate the effectiveness of this programme on the time to recovery in each instrumental ADLs.

As secondary objectives, the effect of this intervention on cognitive function, adherence to the Mediterranean diet, physical function, basic ADLs, symptoms of anxiety and depression, quality of life, anthropometric measures and blood pressure, will be also assessed.

## METHODS AND ANALYSIS
### Study design
The present study is a two-arm parallel, superiority, pilot randomised controlled trial (RCT), which will be nested in the ongoing project 'Multiple Interventions to Prevent Cognitive Decline' (MIND-Matosinhos).[16] This proposed pilot study will provide crucial information about the logistic feasibility, provide estimates of eligibility, recruitment and attrition rates, optimising the study design. It will also enable a greater precision in the power and sample size calculations of a future full-scale study.

The primary endpoints will be the feasibility of this multidomain programme, as well as the time to recovery to pre-event functionality after the occurrence of a TIA or a minor stroke. A block randomisation with a 1:1 allocation will be performed.

### Setting
This pilot RCT will be conducted at the department of neurology of the Hospital Pedro Hispano E.P.E (HPH), as part of the Local Health Unit of Matosinhos (ULSM), which is integrated in the Portuguese National Health Service. The HPH is located at the Matosinhos municipality, Northern Portugal, and is responsible for providing care to 318 000 inhabitants from the municipalities of Matosinhos, Vila do Conde and Póvoa de Varzim.

### Eligibility criteria
Patients aged 18–85 years old and with a clinical diagnosis of TIA or minor stroke as defined by an NIHSS score ≤3 are eligible for this study. Additional inclusion criteria include: the onset of symptoms within the last 7 days; first-time stroke or TIA; at least 4 or more years of education; discharged home without the need for inpatient rehabilitation; modified Rankin Scale (mRS) 0–2. Patients will be excluded if they are unable to attend the face-to-face intervention sessions, or if they have: previous diagnosis of dementia or severe disability; contraindication for physical exercise, severe loss of hearing, vision or communication skills; frailty, reduced life expectancy due to severe disease or need for regular treatments that compete with availability for intervention (eg, chemotherapy, haemodialysis).

### Recruitment
Participants will be enrolled through consecutive sampling from the Neurology Department (stroke or emergency care units) of HPH. Healthcare professionals from the recruitment site will be informed about the protocol and their potential role, and an experienced researcher will be on-site to provide coordination. The neurology medical team will be responsible for screening eligibility and obtaining written informed consent for all the participants. Recruitment started on 11 May 2022.

### Randomisation
The participants will be randomly assigned (1:1) to the intervention or control group after the baseline assessment. Randomisation will be done using a computer-generated programme, using random blocks of four sizes. A researcher who is not immediately engaged in the project will implement the randomisation process.

### Data collection
Participants will be evaluated at baseline (before the beginning of the intervention programme) and at 3 months, as depicted in figure 1. Data on socio-demographic (birth date, sex, address, marital status, education, occupation, income), medical history, current medication, lifestyle (smoking and alcohol use, physical activity and sedentary behaviours) and

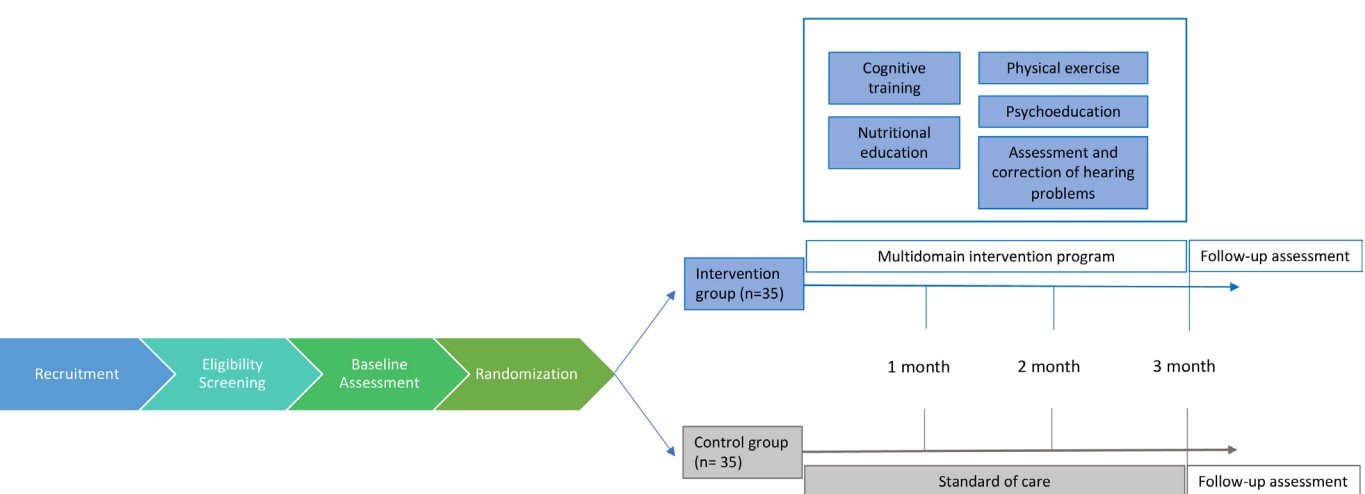

**Figure 1** Study design flowchart

adherence to Mediterranean diet[17] will be collected through structured questionnaires applied by a trained researcher. Anxiety and depression,[18] memory problems[19] and quality of life[20] will be evaluated through self-administered questionnaires validated for the Portuguese population. Additionally, long-term cognition data will be also collected every 3 months for a period of up to 10 years using Brain on Track (BoT), a web-based tool for remote longitudinal assessment of different cognitive domains.[21] The outcome measure section describes the instruments in detail.

Data regarding clinical diagnosis, onset of symptoms, hospital admission and discharge date, mRS score, NIHSS score, medical history of cerebrovascular events, age and need for inpatient/outpatient rehabilitation will be retrieved from the patient's medical record. Clinical characteristics of the cerebrovascular event will also be collected.

## Intervention

The intervention will be implemented over 3 months and will comprise five different components, including cognitive training, physical exercise, nutritional education, psychoeducation and assessment and correction of hearing problems. Only the intervention group will be exposed to all or any of the components of this comprehensive intervention programme. Both groups will continue to receive the usual standard of care provided to this type of clinical disease in the hospital setting. The usual standard of care consists of the initiation of the appropriate secondary prevention and referral to the outpatient cerebrovascular diseases clinic (telephone call at 1-month post-event and in-person appointment at 3 months post-event). Depending on the clinical presentation and neurological deficits, patients can also be referred to an outpatient physical medicine and rehabilitation appointment.

Since early intervention post-vascular event is decisive, given the time interval between neurological injury and the onset of nervous function recovery activities, intervention sessions are aimed to begin within no later than 15 days after the cerebrovascular event. Considering this timeline, once the participants are recruited and randomised, the intervention will start immediately, especially cognitive training.

To minimise refusals and losses to follow-up, all evaluations and sessions will be scheduled by the participant's preference. A written plan with the intervention schedules will be given and sessions reminded through email or phone messages in the day before (day, time and location). If one intervention session is missed, efforts will be placed to arrange another suitable time in another group and individualised meetings will be rescheduled. If not possible, the missed appointment will be documented, and the dropout or non-compliance reasons will be registered for further analysis.

### Cognitive training

Monthly in-person sessions in groups of a minimum of five participants (60 min) will be conducted by a psychologist. Additionally, during the 3 months a daily home-based cognitive training individual (30 min) sessions using the COGWEB[22] online platform will be performed. This online computer tool provides cognitive training tailored to each patient's deficits with different difficulty levels, allowing continuous treatment monitoring, clinical evolution and adherence compliance.[22] Exercises are aimed at different cognitive domains, such as attention, memory, executive function, language and calculus.[22] Participants have access to instructions for each exercise before initiating the session, displayed on the screen and through the speakers. Instructions can be seen at any moment, even throughout the session. Participants without access to a computer will receive paper/pencil equivalent home-based cognitive exercises.

### Physical exercise

Physical exercise sessions in-person or remotely twice a week with a duration of up to 60 min, guided and supervised by a graduated physical exercise technician, will be conducted. Activities will be tailored to each participant tolerance, medical history and comorbidities, focusing on impaired balance and weakness. During exercise sessions,

the exercise technician will track symptoms of chest pain, shortness of breath, wheezing, dyspnoea, lightheadedness, nausea, among others. Small groups allow to set short and attainable goals with the participants and have a graded exercise programme, starting with reduced intensity and gradually increasing.

In-person sessions will be preferred, however if online sessions are needed, the participants without a computer will receive booklets with physical exercise instructions twice a week and the researcher will contact the participants to assess adherence to the exercises.

### Nutritional education

Intervention arm will include individual appointments (three appointments with a nutritionist, once a month with a duration of 45 min to 60 min) and in-person group cooking sessions (once a month with a duration of 90 min). During individual sessions, the nutritionist will emphasise the benefits of adhering to the Mediterranean diet and salt reduction, particularly for overall cardiovascular health advantages. Group sessions start with the nutritionist's presentation of healthy recipes and further analysis of nutritional values, then the participants will undergo its preparation. In these sessions, participants can bring their partners or one family member.

### Psychoeducation

Monthly in-person psychoeducation sessions (60 min) in groups of a minimum of five participants will be conducted by a psychologist. These sessions will support people in understanding, exploring, and self-managing their emotions and impairments, through peer support and promotion of lifestyle changes.

### Assessment and correction of hearing problems

At the beginning of the intervention programme, a hearing assessment and correction will be performed. This evaluation will be conducted by otolaryngologists and audiologists, who will evaluate previous hearing problems and use of hearing aids, and includes otoscopy and audiogram. Those detected with any hearing problem will be referred for a medical appointment with an otolaryngologist.

### Primary outcome measures

The primary outcomes for the study are related with its feasibility and time to partial or complete recovery of each instrumental ADLs.

To evaluate the feasibility of this study, the following primary outcomes will be assessed:

▶ Recruitment time frame: proportion of participants recruited within 7 days after onset of symptoms. It will be assessed by dividing the number of participants recruited within the time frame divided by the total number of participants.

▶ Beginning of the intervention: proportion of participants that started any of the interventions no later than 15 days after onset of symptoms.

▶ Adherence to each component of the intervention: proportion of adherence to each component of the intervention and different intervention modalities (remote/in person), calculated as the number of sessions attended divided by the total number of sessions provided. For the COGWEB training, the outcome will be the absolute number of sessions.

▶ Dropout: proportion of participants who dropped out of the study, calculated as the number of participants who dropped out after attending at least one session divided by the total number of participants who attended at least one session.

Time to recovery in each instrumental activity of daily living will be measured by an adapted version of the Frenchay Activities Index (FAI). The translated and validated version of the FAI—adapted version for the Portuguese population[23] will be used to assess the instrumental ADLs functional performance at baseline and follow-up at 3 months. The version used in this study is based on the perception of the patient's capacity to perform a specific instrumental ADLs instead of its frequency, as stated in the original scale developed by Holbrook and Skilbeck.[24] The scale consists of 15 activities, scored from 1 to 4 (1-could not do at all; 2-very difficult; 3-a little difficult; 4-not difficult at all).[23] Although the original index suggests that total scores for men and women should be considered separately,[24] in the present study the score of each activity will be analysed independently, as the interest in exploring the impact of the programme in each activity.

For the present study, only activities that are part of participants' daily lives will be considered, namely those that were performed at least once in the last 12 months. Time to recovery will be defined as the time from the start of the intervention programme to the time of complete or partial recovery. Each participant will report their level of performance in each activity at the baseline and follow-up at 3 months. In the baseline, if a participant scores a specific activity as altered (score 1–3), their level of performance before the TIA or minor stroke will be confirmed to understand if the change in the activity performance is related with the cerebrovascular event. It will be also inquired and recorded the date when a difficulty in a certain activity ceased. The average of activities altered post-event (score 1, 2, 3) at baseline and the average of activities that improved in both groups on completing the intervention programme will be measured.

### Secondary outcome measures

Secondary outcomes and the instruments that will be used for their assessments are described in tables 1 and 2. Briefly, secondary endpoints related with the study's feasibility will be measured, namely time of follow-up, implemented sessions and complete assessment of participants (table 1). Additionally, cognitive function, quality of life, symptoms of anxiety and depression, as well as functional capacity to perform basic ADLs will be evaluated through questionnaires validated for the Portuguese population

| Table 1 | Description of the secondary feasibility outcomes |
|---|---|
| **Outcome** | **Description** |
| Time of follow-up | Calculated by the number of days between the participant's first and last session attended. |
| Implemented sessions | Proportion of sessions implemented, calculated as the number of sessions that the research team was able to implement divided by the total number of sessions planned. |
| Complete assessment of participants | Proportion of participants with full information at baseline and follow-up points, calculated by dividing the number of participants with complete information by the number of participants evaluated. |

(table 2). Physical condition, blood pressure and anthropometrics will be ascertained by trained interviewers. Levels of glycated haemoglobin, and 24-hour urinary excretion of sodium and potassium will be also measured. The secondary outcomes described on the table 2 will be also measured at baseline and follow-up at 3 months.

### Blinding

Due to the nature of the present intervention, it will not be possible to guarantee a proper blinding to the intervention arm among participants; nevertheless, allocation group will not be actively reported. Regarding the outcomes assessment, interviewers will be independent and blinded to each participants' allocation group.

### Data analysis and sample size

The associations between the social and clinical characteristics and feasibility (eg, recruitment time frame, adherence, dropouts) and lifestyle outcomes will be quantified through linear or logistic regression models, as applicable. Time in days from the onset of symptoms to complete or partial recovery in each instrumental ADLs will be estimated and illustrated by Kaplan-Meier survival curves. The time comparison between intervention and control groups will be made through the log-rank test. To estimate the association between the intervention and the recovery time, HRs will be estimated, through the computation of univariate and multivariate Cox regression models.

Training of the interviewers and use of standardised procedures for data collection is expected to contribute to a low proportion of missing data, and no imputation is being planned.

To the best of our knowledge, there is no previous information regarding effect sizes and SDs regarding the primary outcome related with the recovery of functionality after a TIA or a minor stroke. Nevertheless, a sample size of at least 70 participants (35 per arm) is commonly recommended for a pilot study when estimating the pooled SD.[25] Nevertheless, taking into account the results of a previous RCT evaluating a cardiac model of rehabilitation for reducing risk factors post-TIA and stroke,[26] this sample size will allow to detect mean differences of 5.5 and 6.0 in the changes observed in the Hospital Anxiety and Depression Scale—anxiety and depression scores, respectively, between intervention and control groups (assuming a statistical power of 80% and a level of significance of 5%).

To achieve the sample size needed to accomplish the proposed objectives, we expect to invite approximately 100 individuals, since a participation of 70% was previously observed in the EPIPorto cohort.[27] We also expect a dropout of 10% from baseline to the first follow-up; nevertheless, and in order to minimise refusals and losses to follow-up, all activities will take place near participants' residence and be scheduled for their convenience.

### Contingency plan

This study intents to recruit participants within 7 days after the onset of symptoms and start intervention no later than 15 days, which is a considerably short period. This recruitment time frame may be difficult to attain, so additional measures may have to be adopted, namely receive referrals from other hospitals in neighbouring counties (eg, Porto).

Due to the COVID-19, the number of in-person sessions and participants per group may be reduced. However, if necessary, all activities will be home-based. To minimise the potential impact of the pandemic on participation and retention rates, additional mitigation measures may be adopted, including enlarging the recruitment period or an increase in the sample size.

### Patient and public involvement

Patients and public were not involved in the conception, design and dissemination of this study.

### ETHICS AND DISSEMINATION

Ethics approval was obtained by the Ethics Committee of the ULSM (Ref. N° 75/CES/JAS) and the Data Protection Officer of the ULSM (Ref. 14/CLPSI/2021) and Institute of Public Health of the University of Porto. Any substantive protocol amendment will be reported to the respective Ethics Committee for approval.

Written informed consent will be obtained from all the participants once the study's objectives and procedures are fully explained. The voluntary nature of participation will be emphasised, clearly stating that they can refuse to participate or leave the study with no penalty associated, namely their right to medical care. There are no expected risks or discomfort other than those arising from the sessions, interviews, anthropometric measures and the collection of venous blood and 24-hour urinary samples. The intervention group will be covered by personal accident insurance to indemnify any unlikely accident related

**Table 2** Description of the instruments used for the evaluation of secondary outcomes

| Outcome | Instrument | Description |
|---|---|---|
| Cognitive performance | MoCA[29–31] | Brief screening tool to detect mild cognitive impairment. It assesses eight cognitive domains: visuospatial ability, executive function, attention, concentration, working memory, language, verbal memory and orientation, with scores from 0 to 30. Higher scores represent better cognitive performance, and one point is added for participants with less than 12 years of education. |
| | Brain on Track[21 32] | Self-administered computerised cognitive test for remote longitudinal assessment. It evaluates different cognitive domains through exercises, such as attention, memory, executive functions, language, calculation, constructive capacity and visuospatial processing. The score range is the maximum number of correct answers within a limit of time. Higher scores are representative of better cognitive performance. |
| Memory issues | Subjective Memory Complaints Scale[19 33] | This scale comprises 10 questions regarding subjective issues of memory. This scale varies from 0 (best score) to 21 points (worst score). |
| Adherence to the Mediterranean diet | MEDAS questionnaire[17] | The MEDAS questionnaire is composed by 14 questions associated with eating habits and frequency of food consumption. The score ranges between 0 (lowest adherence to the Mediterranean diet) and 14 points (highest adherence to the Mediterranean diet). A score greater than nine points demonstrates a good adherence to the Mediterranean diet. |
| Anxiety and depression | HADS[18] | Scale with 14 questions, subdivided in two subscales: anxiety (HADS-A) and depression (HADS-D). This scale varies from 0 (best score) to 21 (worst score) for each subscale. The optimal cut-off points differ significantly in the literature. For this study, a score higher than or equal to 11 represents a case of anxiety or depression, as applicable. |
| Reported quality of life | EQ-5D-5L[20 34] | This scale is subdivided into two subscales: (a) five multiple-choice questions, with five response options, which results in a score that fluctuates from 5 (best score) to 25 points (worst score). One digit is provided for each dimension, combining in a 5-digit code. A total of 3125 possible health states are delineated to describe the patient's state of health. (b) Vertical Visual Analogue Scale, which varies from 0 (worst score) to 100 (best score). |
| Disability | Modified Rankin Scale[35 36] | Simple scale that assesses functional capacity, referring to limitations in 'activity and changes in lifestyle'. The scores run from 0 (perfect health) to 6 (dead). It will be considered disability as an mRS of >1 and a one-point change on the scale will be clinically significant. |
| Handgrip strength | Dynamometer[37–39] | Force applied in each hand in kilograms using the JAMAR dynamometer. The handgrip can be a relevant predictor of the physical function of the upper body and influences the capacity to perform ADLs. Participant will be in a seated position with the elbow flexed at 90°. Three alternating hand sequential measurements will be performed and registered. It is relevant to document the dominant hand and the affected and non-affected hands if the case. |
| Agility | TUG[40] | It consists of counting the participant's time to complete the following routine: standing up from a chair, walking for 3 metres (following the marks on the floor), returning to the chair and sitting down as fast as possible. It provides important information about functional capacity, gait, dynamic balance and speed. The time will be counted from when the participant's back leaves the chair until sitting again with their back supported. It will be considered a cut-off level of ≤10 s to identify normal mobility. |
| | Unipedal stance test[41] | This test is performed in time units (seconds) and the score varies from 0 (worse score) to 45 s (best score). The researcher will use a stopwatch and count the amount of time the participant can stand on one leg, of their choice, with their eyes open. |
| | Senior Fitness Test[42 43] | Variation of participant's agility and dynamic balance assessed using the 8-foot distance test (2, 44 metres) from the Senior Fitness Test. |
| Lower limb function | Short Physical Performance Battery[44] | This battery assesses the lower limbs' balance, gait, speed, endurance and strength. The SPPB measures three timed tests: balance in the standing position, walking speed and chair stands. Each test is scored between 0 and 4, total score ranges from 0 (poor performance) to 12 (best performance). |
| Functional fitness | Senior Fitness Test[42 43] | Variation of participant's upper/lower body strength, flexibility and aerobic endurance assessed through the different tests from the Senior Fitness Test. |
| Functional capacity to perform basic ADLs | Barthel Index[45] | Measures functional independence on the basic ADLs. The scores vary from 0 to 20; higher scores represent increased functionality and independence. |

ADLs, activities of daily living; EQ-5D-5L, Measure of health-related quality of life of the EuroQol Group; HADS, Hospital Anxiety and Depression Scale; MEDAS, Mediterranean Diet Adherence Screener Questionnaire; MoCA, Montreal Cognitive Assessment; mRS, modified Rankin Scale; SPPB, Short Physical Performance Battery; TUG, Timed Up and Go test.

to the sessions. Due to the expected low risks and the study small sample size, a data monitoring committee was not established.

To guarantee safety and security of physical exercise, participants with contraindications for exercise practice would not be included in the study. In-person sessions will be preferred, but if sessions are online, some of the exercises will be in a sitting position and participants are encouraged to have someone near them throughout the sessions at home.

This study involves collecting, analysing and processing sensitive personal data, including contact information, health and clinical data from questionnaires and participants' clinical records. Therefore, measures are in place to guarantee

the confidentiality and anonymity of the participants. All data from clinical records will be collected by clinical members of the research team and privacy will be assured. Health and clinical data will be pseudonymised, with each participant being assigned and identified with a unique numerical code in the study. The principal investigator will be the only person with access to the file containing the correspondence between this unique numerical code and the personally identifiable information. Only specific members of the research team have access to the names and contact information of the participants, which will be only used to book evaluations and intervention sessions and resolve any logistical problem throughout the implementation of the intervention programme.

Documents with personal identification data and informed consents will be securely stored, separated from other documents and with access limited to the research team and no personal identifiers will be used in data analyses. Only the research team will have access to the study data set, saved on a password-protected secure computer. The study is in accordance with the Helsinki Declaration of 1964, further amendments[28] and applicable national and European union legislation.

The implementation of the present study will elucidate the feasibility and impact of a multidomain non-pharmacological programme in recovering functionality after a TIA or a minor stroke, which may contribute to delineate further larger RCT and adequate preventive actions. This study will also provide important data to the refinement and validation of the BoT, which will allow a more practical, intensive and economic monitoring of cognitive performance in this specific population. The expected results will be relevant for public health policies, by providing a characterisation of residual sequelae of a TIA or a minor stroke, to improve patients' functionality and promote a faster recovery. It may, ultimately, contribute to a better allocation of resources and the creation of synergies between different stakeholders, as well as to improve the management of care.

The findings of this study will be disseminated at several levels, including the publication of scientific papers in international peer-reviewed journals and presentation in relevant national and international conferences. A short training course on secondary prevention post-stroke for public health researchers and students will be also conducted and the project will contribute to the training of researchers through the production of a master dissertation. The main conclusions of the project will be disseminated through press releases and conferences to the media (including newspapers, television and social media such as Facebook, LinkedIn and Twitter), fostering a connection between science and society.

**Author affiliations**
[1]EPIUnit, Instituto de Saúde Pública da Universidade do Porto, Porto, Portugal
[2]Laboratório para a Investigação Integrativa e Translacional em Saúde Populacional (ITR), Porto, Portugal
[3]Serviço de Neurologia, Unidade Local de Saúde de Matosinhos EPE, Matosinhos, Portugal
[4]Departamento de Ciências da Saúde Pública e Forenses e Educação Médica, Faculdade de Medicina da Universidade do Porto, Porto, Portugal

**Contributors** MG and VTC conceived and designed the study. MG wrote the first version of the manuscript. MJL, ARC, AF and CD collaborated in refining study design and critically revised the manuscript for relevant intellectual content. All authors approved the final manuscript as submitted.

**Funding** This work was financed by national funds through the FCT - Foundation for Science and Technology, I.P., within the scope of projects UIDB/04750/2020 and LA/P/0064/2020. It was also supported by the Portuguese Stroke Society under the research scholarship Prof. Castro Lopes in cerebrovascular disease. The "MIND-Matosinhos" project was supported by 'Portugal Inovação Social' and co-funded by the Operational Program Social Inclusion and Employment, Portugal 2020 and European Union, through the European Social Fund (POISE-03-4639-FSE-000793). Funders will not have any influence on the study design, execution, interpretation of data, writing and publication of manuscripts.

**Competing interests** VTC has a shareholder position in Neuroinova, Lda, a start-up company that conceived Brain on Track and holds registered trademark and commercialisation rights.

**Patient and public involvement** Patients and/or the public were not involved in the design, or conduct, or reporting, or dissemination plans of this research.

**Patient consent for publication** Not applicable.

**Provenance and peer review** Not commissioned; externally peer reviewed.

**ORCID iD**
Micaela Gonçalves http://orcid.org/0000-0001-6784-4871

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
