## [Reviewer comments · BMJ Open]

ARTICLE DETAILS

TITLE (PROVISIONAL)	A study protocol for a pilot randomised controlled trial evaluating the feasibility and effectiveness of non-pharmacological interventions to recover functionality after a transient ischemic attack or a minor stroke: the "Back to Normal" trial
AUTHORS	Gonçalves, Micaela; Lima, Maria João; Fonseca, Ângelo; Duque, Cristina; Costa, Ana; Cruz, Vitor

VERSION 1 – REVIEW

REVIEWER	Wang, Suzie Leeds Beckett University, Psychology/Humanities and Social Sciences
REVIEW RETURNED	04-Jan-2023

GENERAL COMMENTS	This is a well written protocol of a study which meets important clinical needs of people who have had a TIA or minor stroke. I just have some queries that would benefit from the authors' clarification, please. Thank you. - Ls 2-5, p. 10 how will the eligibility criteria of 'severe loss of hearing, vision, or communication skills; frailty' determined /measured? What threshold(s), if applicable, constitute as 'severe'?- Ls 21-23, p. 15, it is a great idea to tailor physical activities to "each group's tolerance, medical history, and comorbidities". Would you clarify how you make sure each group has similar level of tolerance, medical history, and comorbidities? Or is this not necessary?- Ls 33-37, p. 15, do the authors expect a performance and/or adherence difference of participants who take part in-person vs those who do it online? Will this difference affect the outcomes? If no difference is expected, what are the rationale/evidence that these two different formats of delivery will yield similar results?- Ls 6, p14, is the psychoeducation also conducted in a group setting?- Will the secondary outcome measures listed in Table 2 also measured at baseline so to detect any differences among them pre and post follow up?
--

REVIEWER	Rudd, Anthony Royal College of Physicians, London, Clinical Effectiveness and Evaluation Unit
-----------------	--

REVIEW RETURNED	18-Jan-2023
-------------

GENERAL COMMENTS	The paper describes the methodology used for a pilot trial of multifaceted rehabilitation after TIA and minor stroke including dietary, cognitive, psychological and physical rehabilitation performed both on line and face to face. This is an area of considerable interest although most of the individual components have been evaluated before. I would look forward to seeing the results of a full scale trial. My comments are:  1. The pilot study completed data collection in December 2022 with final follow up 3 months later so it is quite late to be planning to publish the protocol. 2. Clearly it is too late to influence the design of the study but for this sort of intervention I would have thought that one of the key factors to assess would be the longer term compliance with follow-up. In the present study the intervention goes on for 3 months and follow up data also at the 3 month timepoint. Given that the interventions are designed to have a long term impact it is a shame that the ability to collect the longer term data is not tested 3. It is stated that the assessors for the outcome measures would be blinded. How will this be achieved and is there any measure included to assess how effective blinding had been? 4. Under the section on strengths and limitations, I found the sentence - 'Given the characteristics of the intervention, blinding of participants is not possible, the main outcome is based on a pre-defined list of activities of daily living and will be self reported, being prone to recall bias' - difficult to understand. Which measure is prone to recall bias?
--

VERSION 1 – AUTHOR RESPONSE

Reviewer: 1
Dr. Suzie Wang, Leeds Beckett University

This is a well written protocol of a study which meets important clinical needs of people who have had a TIA or minor stroke. I just have some queries that would benefit from the authors' clarification, please. Thank you.

We thank the reviewer for the positive comment. No specific questions raised.

Ls 2-5, p. 10 how will the eligibility criteria of 'severe loss of hearing, vision, or communication skills; frailty' determined /measured? What threshold(s), if applicable, constitute as 'severe'?

In this pilot study, we do not consider specific thresholds or scales to determine the severe loss of hearing, vision or communication skills, as well as frailty. These criteria are verified by a clinical assessment performed by the neurology medical team on the primer evaluation of the eligibility of potential participants. Severe loss of hearing, vision or communication skills implies a condition that

would not allow people to complete the baseline and final assessments, along with the intervention program, given that proper accommodations are not available in this pilot study for such states.

Regarding specifically the evaluation of frailty, we acknowledge the importance of assessing pre-stroke frailty using available measures, such as the Clinical Frailty Scale or Fried frailty phenotype [1]. In fact, despite not being done in this pilot study, due to its prognostic value it will be considered when designing a potential full trial in the future.

- Ls 21-23, p. 15, it is a great idea to tailor physical activities to “each group's tolerance, medical history, and comorbidities”. Would you clarify how you make sure each group has similar level of tolerance, medical history, and comorbidities? Or is this not necessary?

Regarding physical activity, we strive to have homogeneous groups, and it is possible to shift participants' allocation to a more or less intensive group sessions according to one's exercise tolerance that can be a reflection of certain medical conditions. However, this might not always be possible, due to participant's schedule limitations. Additionally, the physical exercise technician is informed about the medical history and comorbidities that may affect the participant's performance throughout the session and takes that into consideration when planning the class. The response of the participant throughout the session is evaluated, and if deemed necessary, the exercise technician suggests a change to a different group which will be more suitable. Furthermore, all exercises are tailored to each participant's condition, which might mean less repetitions, seating position exercises, bodyweight or weight-lifting training, or other modifications to engage every participant. Nevertheless, in order to clarify this topic, we amended the respective sentence in the methods section, referring that “*Activities will be tailored to each participant tolerance, medical history, and comorbidities*” (please see page 11, paragraph 2).

- Ls 33-37, p. 15, do the authors expect a performance and/or adherence difference of participants who take part in-person vs those who do it online? Will this difference affect the outcomes? If no difference is expected, what are the rationale/evidence that these two different formats of delivery will yield similar results?

Previous studies have shown that online lifestyle programs might be as effective and feasible as in-person sessions [2,3]. However, these results should be taken with caution given the novelty and the scarce high-quality available research comparing both ways of delivering interventions. In this context, we are expecting similar performance and/or adherence between the participants who take part in-person and those who do it online, but this cannot be deduced to a high degree of certainty.

In our study, the online format was designed to reduce barriers to participation, either related with patient preference or public restrictions related with the COVID-19 pandemic. Furthermore, if the online mode is preferred, participants have synchronized sessions with the professionals, which is an important factor to promote adherence and the overall impact of the online intervention sessions.

- Ls 6, p14, is the psychoeducation also conducted in a group setting?

Yes, the psychoeducation session is conducted in a group setting. We agree with the reviewer and decided to amend the sentence to make it clearer: *“Monthly in-person psychoeducation sessions (60 minutes) in groups of a minimum of five participants will be conducted by a psychologist. These sessions will support people in understanding, exploring, and self-managing their emotions and impairments, through peer support, and promotion of lifestyle changes.”* (please see page 12, paragraph 1).

- Will the secondary outcome measures listed in Table 2 also measured at baseline so to detect any differences among them pre and post follow up?

Yes, we confirm that the secondary outcomes will be measured at baseline and then at the final evaluation, after three months. Moreover, we acknowledge that it was not clear on the initial manuscript, so an update has been made to the new version:

“The secondary outcomes described on the Table 2 will be also measured at baseline and follow-up at three months” (please see page 14, paragraph 2).

Reviewer: 2

Dr. Anthony Rudd, Royal College of Physicians, London

Comments to the Author:

The paper describes the methodology used for a pilot trial of multifaceted rehabilitation after TIA and minor stroke including dietary, cognitive, psychological and physical rehabilitation performed both on line and face to face. This is an area of considerable interest although most of the individual components have been evaluated before. I would look forward to seeing the results of a full scale trial.

We thank the reviewer for the positive comments. No specific question raised.

1. The pilot study completed data collection in December 2022 with final follow up 3 months later so it is quite late to be planning to publish the protocol.

We agree with the reviewer that the study would have benefited from an early publication of the protocol. However, we would like to inform that the recruitment process was extended until March 2023 and the data collection process is expected to be completed by July 2023. As such, the new version of the manuscript will reflect that update (please see page 2, paragraph 2). Nonetheless, the publication of

this study protocol might help in designing, refining and improving the current methodology for a potential full-scale multicenter trial and informing the scientific community of the current research.

2. Clearly it is too late to influence the design of the study but for this sort of intervention I would have thought that one of the key factors to assess would be the longer term compliance with follow-up. In the present study the intervention goes on for 3 months and follow up data also at the 3 month timepoint. Given that the interventions are designed to have a long term impact it is a shame that the ability to collect the longer term data is not tested.

As referred by the reviewer, we recognize the importance of collecting long-term data to assess the lasting impact of the interventions, and if the opportunity arises, this would be considered when designing a full-scale trial. However, despite its relevance, the intention of this pilot study is not to answer that question, but rather to identify potential refinements in the intervention program, address potential issues around the feasibility of the recruitment process, delivery of interventions, adherence to each component and appropriateness of the primary outcome instrument measure.

3. It is stated that the assessors for the outcome measures would be blinded. How will this be achieved and is there any measure included to assess how effective blinding had been?

The outcome measures are evaluated by independent assessors who are not informed about the participant's group allocation. Participants are requested to not mention their allocation status throughout the evaluation, but we cannot guarantee that this is achieved in all the cases. Nevertheless, since this is an important issue, we will include a formal question to the independent assessors to evaluate if they were aware of participants' group allocation at the end of the final evaluation.

4. Under the section on strengths and limitations, I found the sentence - 'Given the characteristics of the intervention, blinding of participants is not possible, the main outcome is based on a pre-defined list of activities of daily living and will be self reported, being prone to recall bias' - difficult to understand. Which measure is prone to recall bias?

The measure prone to recall bias is the Frenchay Activities Index, which is a self-reported instrument. We agree with the reviewer that the sentence was unclear and was revised: *"Given the characteristics of the intervention, blinding of the participants is not possible. In addition, the main outcome is evaluated by self-report and is based on a pre-defined list of activities of daily living. Therefore, we cannot exclude the existence of recall bias regarding our main outcome"*. (please see page 4, fifth bullet point).

References

- 1 Burton JK, Stewart J, Blair M, et al. Prevalence and implications of frailty in acute stroke: systematic review & meta-analysis. Age Ageing 2022;51:1–10.
- 2 Villegas V, Shah A, Manson JAE, et al. Prevention of type 2 diabetes through remotely-administered lifestyle programs: A systematic review. Contemp Clin Trials 2022;119.
- 3 Brown RC, Coombes JS, Rodriguez KJ, et al. Effectiveness of exercise via telehealth for chronic disease: a systematic review and meta-analysis of exercise interventions delivered via videoconferencing. Br J Sports Med 2022;56:1042–52.

VERSION 2 – REVIEW

REVIEWER	Wang, Suzie Leeds Beckett University, Psychology/Humanities and Social Sciences
REVIEW RETURNED	19-Mar-2023

GENERAL COMMENTS	Thank you for the clarification and revision from the authors. I think the manuscript is strengthened and improved and am happy to accept this revised version.
---

REVIEWER	Rudd, Anthony Royal College of Physicians, London, Clinical Effectiveness and Evaluation Unit
REVIEW RETURNED	13-Mar-2023

GENERAL COMMENTS	I thanks the authors for addressing the issues raised by me and my co reviewer. I am happy with the responses and have no additional comments to make.
--